# “If I Was Educated, I Would Call the Ambulance and Give Birth at the Health Facility”—A Qualitative Exploratory Study of Inequities in the Utilization of Maternal, Newborn, and Child Health Services in Northern Ethiopia

**DOI:** 10.3390/ijerph191811633

**Published:** 2022-09-15

**Authors:** Alem Desta Wuneh, Afework Mulugeta Bezabih, Lars Åke Persson, Yemisrach Behailu Okwaraji, Araya Abrha Medhanyie

**Affiliations:** 1School of Public Health, College of Health Sciences, Mekelle University, Mekelle P.O. Box 1871, Ethiopia; 2Department of Disease Control, Faculty of Infectious and Tropical Diseases, London School of Hygiene & Tropical Medicine, London WC1E 7HT, UK

**Keywords:** maternal and child health services, utilization, inequity, qualitative, Ethiopia

## Abstract

In earlier studies, we have shown that the utilization of maternal health services in rural Ethiopia was distributed in a pro-rich fashion, while the coverage of child immunization was equitably distributed. Hence, this study aimed to explore mothers’ and primary healthcare workers’ perceptions of inequities in maternal, newborn, and child health services in rural Ethiopia, along with the factors that could influence such differentials. A qualitative study was conducted from November to December 2019 in two rural districts in Tigray, Ethiopia. Twenty-two in-depth interviews and three focus group discussions were carried out with mothers who had given birth during the last year before the survey. We also interviewed women’s development group leaders, health extension workers, and health workers. The final sample was determined based on the principle of saturation. The interviews and focus group discussions were audiotaped, transcribed, translated, coded, and analyzed using thematic analysis. Two major themes emerged during the analysis that characterized the distribution of the service utilization and perceived causes of inequity. These were: (1) perceptions of the inequity in the use of maternal and child health services, and (2) perceived causes of inequity in maternal and child health service utilization. The mothers perceived antenatal care, facility-based delivery, and care-seeking for sick children to be inequitably distributed, while immunization was recognized as an equitable service. The inequity in the maternal and child health services was linked to poverty, lack of education, lack of access, and poor-quality services. The poor, the uneducated, and women who were distant from health facilities had a low utilization rate of services. The weak implementation of community-based equity-oriented policies, such as community-based health insurance, was perceived to result in health inequities. Mothers and primary healthcare providers in rural Ethiopia experienced weaknesses in delivering equitable services. The narratives could inform efforts to provide universal health coverage for mothers, newborns, and children by improving access and empowering women through poverty alleviation and education.

## 1. Introduction

Enhancing equitable maternal, neonatal, and child health services increases the possibilities to reach the current global goals of universal health coverage [1]. Equity was not an identified goal or target in the Millennium Development Goals [2]. Despite progress in the coverage of health services and maternal and child health indicators, inequities remain a challenge [3,4]. In response to this problem, equity was prioritized in the Sustainable Development Goals (SDGs) [5]. Inequities in the utilization of maternal health services in Africa have been linked to household wealth, a proxy for poverty, the distance between home and health facilities, parents’ educational level, maternal age, and rural or urban residence [6]. Social differences and gender inequities have led to the low utilization of maternal health services [7]. Efforts to reach disadvantaged groups with maternal, newborn, and child health services need to be guided by knowledge of the coverage and different inequality dimensions [4].

In Ethiopia, antenatal care is provided at health posts and health centers, and skilled assistance at delivery is freely available at health centers and hospitals. Child immunization services are provided through outreach services and campaigns, in addition to the immunization services and sick child services that are available at health facilities. All pregnant women are recommended to receive at least four antenatal care visits [8]. As a result, Ethiopia has made substantial progress in maternal, newborn, and child health in recent decades [9]. The proportion of pregnant women attending antenatal services four or more times increased from 12% in the 2005 Demographic and Health Survey to 43% in 2019. Over the same years, facility-based deliveries increased from 5% to 48%. The under-five mortality rate was 123 per 1000 live births in the 2005 survey, and 55 per 1000 in 2019. The neonatal mortality rate changed less, from 39 per 1000 live births in the 2005 study to 30 per 1000 live births in 2019 [10]. Despite this progress, inequities in service coverage and different maternal and child health outcomes remain a challenge [11]. The coverage of maternal health services and basic child immunizations has favored the wealthiest, the more educated, and those living in urban areas [10]. The Ethiopian Government has committed itself to improve equity through the health extension program and other initiatives [12]. The equity focus is relatively new in Ethiopia, and was included in the health sector transformation plan [13] after the Millennium Development Goal era. This perspective was prompted by the large social and geographical differences in the coverage of health services [14].

In a recent survey in four regions in Ethiopia, we found that maternal health service utilization was low and inequitably distributed, favoring the better-off mothers. In contrast, basic child immunization coverage was low but equitably distributed [15]. These results prompted us to explore why the maternal health services were inequitably distributed whilst the immunization services equitably distributed. We also aimed to explore mothers’ and primary healthcare workers’ perceptions of inequity, and why inequities existed in the provision and utilization of maternal, neonatal and child health services in rural areas of northern Ethiopia. We were guided by the conceptual framework on social determinants of health proposed by the World Health Organization [16].

## 2. Study Setting

This study was conducted in two districts of the Tigray region; Enderta and Saharti Samre. The Ethiopian healthcare system is three-tiered, comprising primary, secondary, and tertiary care. The health system is responsible for providing maternal and child health services, such as antenatal care, facility-based delivery, postnatal care, immunizations, and the treatment of sick children. The primary care unit includes primary hospitals, health centers, and health posts which are responsible for providing services for the rural population [17]. The health extension workers (HEWs) are supported by women’s development groups (WDGs) that mobilize and link women to health facilities.

### 2.1. Study Design, Sampling, and Recruitment

This was an exploratory, descriptive, qualitative study [18]. We recruited the study subjects using purposive sampling. We selected four *tabias* (the lowest administrative unit) from the two mentioned districts, in consultation with the head of the district health office. Two tabias were far away and close to the health center. The participants in the in-depth interviews (22 women) and focus group discussions (27 women) had given birth during the year before the interview. WDG leaders and healthcare workers at different levels were also interviewed. The participating mothers were recruited by the HEWs and WDG leaders. The health workers responsible for the maternal, neonatal and child health services working at the health facilities in the districts were included in the study. They were recruited by the recommendation of their immediate supervisors. We collected our sample based on the principle of saturation; data collection was completed when no new information was generated [19].

### 2.2. Data Collection

The data were collected between November and December 2019. Three university instructors and researchers with master’s degrees who had prior experience in qualitative data collection and analysis, together with the principal investigator, collected the data after one day of training. The interviews and focus group discussions were undertaken using a pretested guide prepared in English Appendix A, and were later translated into the local language, Tigrigna. The field guides were developed based on the Commission on Social Determinants of Health (CSDH) conceptual framework on the social determinants of health [16]. We selected this conceptual framework because of its appropriateness and wide use in studies of health inequity. The guide included questions on maternal, newborn, and child health service utilization; any perceived difference between groups in the utilization of services; the mothers’ experiences of services; and perceived causes of inequity or social determinants in maternal, newborn, and child health service utilization. In the process of data collection, the field guides were continuously reviewed and improved.

The in-depth interviews with mothers were carried out face-to-face in a separate room in their private homes, and with healthcare workers in their private rooms in order to ensure privacy. The focus group discussions with mothers were held at a nearby health post. The interviews and discussions were conducted in the local language, Tigrigna.

During the focus group discussions and interviews, we established rapport in order to make participants feel comfortable to trustworthily share their perceptions and experiences. The interviewers briefed the participants about the study’s purpose and the confidentiality of their contributions. After providing this information, the interviewers obtained written informed consent. All of the in-depth interviews and focus group discussions were audiotaped, transcribed, translated and kept available only to the involved researchers. The interviewers met daily with the principal investigator to discuss emerging information, refine the guides, and ensure consistency in the data collection. There was peer debriefing throughout the data collection and analysis in order to ensure the validity of the data. Member checks were also carried out with the health workers. Information collected through different qualitative methods was triangulated in order to enhance credibility.

### 2.3. Data Analysis

The data were analyzed using thematic analysis [18] according to the theoretical approaches of Miles and Huberman [20]. We followed a step-by-step analytic strategy; the transcribed data were coded, thematized, and complemented with hand-written field notes. The data collectors performed the translation and transcription of the interviews and focus group discussions. The transcription was carried out word-for-word in order to capture all of the voices, and was reviewed repeatedly in order for us to become familiarized with the data. The principal investigator and the data collectors analyzed the data through discussion, i.e., any disagreements were solved through negotiation. The analysis started during the data collection. This start allowed us to know the nature of the data and determine whether saturation had been reached. Themes were formed by grouping codes that had similar concepts. The thematic analysis was guided by the conceptual framework for action on the social determinant of health [16]. The themes were supplemented by verbatim quotations to present the respondents’ perspectives. Atlas’ qualitative software (version 7.5.16) was used to store and analyze the data. The coding was reviewed repeatedly, and revisions were made accordingly.

**Regarding ethical clearance**, The Institutional Review Board of the College of Health Sciences at Mekelle University, Ethiopia, approved the study (Ref. No.: ERC 1434/2018). Informed and written consent were obtained from all of the interviewees and focus group discussion participants. All of the respondents were informed that their participation in the study was voluntary, and that the data would be stored safely, without identifiers, and would only be accessed by the involved researchers.

## 3. Results

### 3.1. Characteristics of the In-Depth Interviewees and Focus Group Participants

The age of the interviewed women varied from 22 to 40 years. A majority had a primary (grades 1–8) level of education. Most of the focus group participants were between 25 and 34 years old. About half of them had attended primary education. The healthcare providers were two HEWs and four health workers. Four WDG leaders also participated in the study. The interviews, as well as the focus group discussions, had a duration of 45 min to two hours.

Two major themes emerged from the data based on the participants’ sayings that characterized the use of maternal, neonatal, and child health services. These themes were (1) *perceptions about inequity in the use of maternal, neonatal and child health services*, and (2) causes *of inequity as perceived by the informants*. The causes of inequity were classified into five sub-themes, which were (1) socioeconomic factors, (2) lack of access, (3) perceived poor quality of care, (4) inappropriate implementation of equity-oriented policy initiatives, and (5) suggestions to improve equity in the utilization of health services (Table 1).

### 3.2. Perceptions about Inequity in the Use of Maternal and Child Health Services

Antenatal care and delivery at health facilities were perceived as the most inequitably distributed services. Most of the interviewed mothers expressed that antenatal care and facility-based delivery were not utilized appropriately. The WDA leaders considered antenatal care to be problematic. Similarly, the health workers reported that some mothers did not come to health facilities for delivery. These mothers were reported to experience home deliveries. Home delivery was common among poor women from remote areas.

The interviewed health workers reported that postnatal care for mothers and their babies had very low coverage. The interviewed mothers also mentioned that once a woman had delivered safely, she did not see the importance of going back to the health facility for a check-up. The women associated postnatal visits with illness or complications that happened to them or their children.


*After birth, I was healthy. I did not see any reason to visit the health center for a checkup. We (mothers) go there if we or our babies feel unhealthy.*
(In-depth interview, woman, age 30 years)

A young HEW said that *mothers visit health posts for vaccination only, but not for a health check-up*. Another healthcare provider also noted that mothers who gave birth at the health facility did not return for subsequent postnatal care visits.

However, all of the participants perceived that child immunization was fairly distributed. The interviewed mothers unanimously expressed that they got their children vaccinated. It is the most accepted service, all of the interviewed mothers said. The HEWs, supported by the WDGs, provided the immunization services in every village once a month, marked on a religious day. Equity in immunization was associated with improved access to the service that was provided through outreach activities on a convenient day to the mothers.


*There is no single woman remains at home during the vaccination day. Vaccination is the most acceptable service for all mothers in our village. They (health extension workers) come to our village every month on the day of St. Mary, our rest day, and immunize our children.*
(In-depth interview, woman, age 35 years)

### 3.3. Perceived Causes of Inequity in the Use of Maternal and Child Health Services

#### 3.3.1. Socioeconomic Factors

Two factors were identified under the socioeconomic factors which were perceived to cause inequities in maternal and child health service utilization: lack of economic resources and women’s education.

Most of the participants linked the difference in service utilization to a lack of economic resources. Poor mothers did not visit health facilities for maternal and child health services. They mentioned that money was needed for transport. The transportation cost for escorts was also an additional burden to the poor. Poor mothers could not afford to buy medicines from private pharmacies.


*We (the poor) do not have money to pay for transportation and we are giving birth at home, [though we know that the services are free]. We want to give birth at the health facility but it is the lack of money that is deterring us from going there.*
(In-depth interview, woman, age 22 years)

The interviewed health workers and HEWs also acknowledged the influence of the economy, which hindered poor mothers from utilizing the maternal and child health services.


*Women from wealthier households hire minibuses with 300 Ethiopian Birr (USD 8.1) and are delivering at the health facility. However, poor women don’t have such opportunities.*
(In-depth interview, health extension worker, age 28 years)

Additionally, the participants repeatedly mentioned that money made it a challenge for poor women to buy food and medicines for sick children. This was even worsened when the women were told to buy drugs from private pharmacies.


*The poor woman does not seek care at the health facilities when her child gets sick. This is because she does not have money to buy medicines. [So, what can a poor woman do if she doesn’t have money at hand? She can do nothing at all]. She cannot even get a loan.*
(In-depth interview, woman, age 35 years)

Education was perceived as an important factor that caused inequity in the utilization of maternal health services. The participants indicated that uneducated women were less likely to use health facilities for antenatal care and delivery. They associated this with less knowledge and understanding of the benefits of utilizing maternal, neonatal, and child health services.

Moreover, the interviewed uneducated mothers expressed that they had difficulties in making phone calls to ambulance drivers. They reasoned that they did not know how to make phone calls. As a result, they gave birth at home.


*The health extension workers provided me with the phone number of the ambulance drivers. However, I did not make a phone call because I am not educated. If I was educated, I would call the ambulance and give birth at the health facility.*
(In-depth interview, woman, age 30)

In contrast, some of the respondents argued that the younger mothers utilized the health facilities better than the older ones. They associated this with education; the younger mothers today were more educated than the older ones. These mothers had better knowledge of the benefits of utilizing maternal health services. An interviewed WDG expressed her view on the younger mothers’ health-seeking practices as follows:


*Most young mothers today are educated. There are even some mothers who completed grade 10. They have better understanding of the benefits of health services. As a result, they (young mothers) attend antenatal care better and deliver at health facilities more often than the older ones.*
(In-depth interview, Women’s development group leader, age 58 years)

#### 3.3.2. Lack of Access

Distance to health facilities was perceived as a deterring factor for accessing the health facilities by almost all of the participants. The distance was complemented by poor road conditions, mountainous topography, and a lack of transport that negatively impacted maternal health service utilization. Most participants mentioned that mothers from remote areas were most affected. Some participants complained that they walked for up to three hours to reach the nearest health center.


*The health center is too far away from our home. In our village, let alone a pregnant woman, it is even more difficult for a non-pregnant woman to visit the health facility and seek medication. For example, it took me about three hours to the health center for antenatal care follow-up, six hours, including the back trip. It would be impossible to consider and fully attend antenatal care visits.*
(Focus group discussion, woman, age 40 years)

Traditional stretchers were previously used to transport pregnant women to health facilities for delivery. However, the women were not utilizing them due to drivers’ uncooperative and unfriendly attitude. They expressed that the drivers barely respond to phone calls, or put their cell phones off. Consequently, poor women from remote areas were experiencing home births. Some gave birth on the road when traveling to the health facilities because of delays in the ambulance.

The interviewed health workers echoed the mothers’ opinions that long distances prevented women from going to health facilities for antenatal care and delivery. The health workers expressed that mothers from remote areas hardly ever completed the recommended number of antenatal care visits. The participants, therefore, suggested connecting villages to health facilities with roads in order to improve transportation access.

Additionally, sociocultural factors, such as a lack of husbands’ support, mothers’ cultural taboos, and mothers’ heavy housework were also perceived to influence the mothers’ health-seeking behavior, especially the poor ones. Some participants expressed that husbands did not support their wives during pregnancy. They migrated to urban areas in search of jobs, leaving their wives without support.


*I gave birth at home because I was alone. My husband was away. He moved to Ketama (an urban area) to bring food to our family.*
(In-depth interview, woman, age 35 years)

Some women said that some husbands were helpful. These husbands supported their wives; they accompanied them while traveling to health facilities for delivery. Some also stayed at home caring for their children.


*Our husbands encourage us to visit the health center. I showed my appointment card to my husband and he said: you need to go to the health center; I will take care of my children and animals.*
(Focus group discussion, woman, age 27 years)

The respondents suggested that husbands need to stay at home when women become pregnant. This can enhance their attendance to antenatal care. They also said that husbands should take part in household responsibilities and arrange transport for laboring women to deliver at health facilities. It was highlighted that the husbands’ lack of support reduced their wives’ health service utilization. The WDG leaders described that husbands’ primary responsibility was to make money for their families, with them sometimes traveling far away from home. Although husbands generated household income, their absence negatively influenced their wives’ health-seeking behavior. It was noted that the lack of husbands’ support was common among poor families.

The busy work of women with many household chores also reduced their ability to attend antenatal care.


*We (pregnant women) do not have spare time during pregnancy. We work until the end of our pregnancy- cooking, caring for our children, and cleaning the house. We are very busy with our housework. We do not have time to visit the health center.*
(In-depth interview, women’s development group leader, age 30 years)

The informants perceived that that cultural norms limit post-delivery care-seeking. In the local tradition, women were restricted from moving outside of their homes after delivery. This constrained mothers from attending postnatal care until the time their children got baptized. They also had a fear of exposing their babies to the evil eye and witchcraft when attending postnatal care.


*It is uncommon to attend the services after delivery. In our village, we (mothers) do not attend postnatal care at the nearby health center. It is not customary. A mother is not allowed us to go out after delivery before baptizing her babies.*
(In-depth interview, woman, age 35 years)

Moreover, a health worker mentioned that low education and a lack of money, combined with the distance to the facility, were the main reasons for poor women not to use the services. The health worker illustrated his view as follows:


*Mothers have multifaceted problems that distance them from the health facilities. First, household-related problems such as lack of money; second, lack of awareness and understanding of the benefits of utilizing the health facilities; third, limited access to transportation. The co-existence of these problems widens the distance between home and health facilities. If you are far away from the health facility, you are also away from information. If you do not have money, you are unable to get transportation. Thus, limited economic resources, lack of access to information, and no transportation inflate the distance to the health facilities.*
(Health worker, age 26 years)

#### 3.3.3. Perceived Poor Quality of Care

The participants had a range of experiences with health workers’ behavior at the health centers. Some had been welcomed and treated with respect, while others had been mistreated. The informants complained of the health workers’ unreasonable annoyance when a mother arrived late, gave birth on the way to health facilities, or gave birth at home due to the delay or absence of an ambulance. They mentioned that the health workers did not listen or understand that the situation was outside of their control. They also stated that health workers’ bad behavior discouraged mothers from utilizing the health facilities.


*I have seen a health care provider snapping at a laboring woman. They had to refer her to Mekelle, the regional referral hospital, and then she cried. They (health workers) then said you did not feel shame when you got pregnant, but you lost the shame and cried while giving birth. They were just joking. Also, another female hakim (health worker) came and snapped at the crying woman. It has never happened to me, but what I observed discourages women to utilize the health facilities.*
(Focus group discussion, woman, age 30 years)

Many of the focus group discussants stressed that health workers’ bad behavior negatively affected women’s care-seeking behavior. It was highlighted that negative experiences have spillover effects on other women’s health-seeking behavior.

In contrast, a woman assisted by the health workers during delivery at a health center described the staff as very respectful and caring.


*The health workers who assisted me during delivery were respectful and caring. Even your mother can’t do what the health workers did for me. Today, they (health workers) take the role of our parents in caring for us. They are much worried and highly concerned about your health. For example, I had experienced bleeding while giving birth, but they immediately injected me with a drug and stopped the bleeding. I thank them all for saving my life.*
(In-depth interview, woman, age 26 years)

Similarly, a WDG leader expressed the health workers’ fairness and non-discriminating treatment of women based on her own lived experiences:


*The health workers are caring, especially for poor mothers like me. I am poor and was referred to Mekelle regional referral hospital. I found the health workers there very caring. Some patients dressed neatly and had their bedsheets and blankets. But I was served equally with those patients. I slept at the hospital for about three weeks, and I was satisfied with the services they provided me.*
(In-depth interview, women’s development group leader, age 39 years)

All of the focus group participants and most in-depth interviewees explained that the unavailability of medicines at government health facilities was a significant concern. The informants stated that they were told to buy medicines from private pharmacies. This lack of medicines reportedly deterred poor women from using health facilities. They also expressed that they lacked trust because of the stockout of medicines.


*I brought my child to the health center. The healthcare workers examined him and wrote a prescription and told me to buy drugs from private pharmacies. I feel distrust when medicines are lacking at government health facilities. I went back home without having the drugs. How can a private pharmacy have a better supply of drugs than government facilities?*
(Focus group discussion, woman, age 40 years)

#### 3.3.4. Inappropriate Implementation of Equity-Oriented Policy Initiatives

The informants acknowledged that the government has developed several community-based initiatives to facilitate the equitable use of maternal and child health services. Nonetheless, the informants also stated that they were not satisfied with the health insurance and ambulance services. Health insurance was designed to address the financial barriers of the poor. However, the informants reported that they were not benefiting from it. The focus group discussants expressed that the health insurance scheme was becoming a reason for the unavailability of medicines. They said that a patient with an insurance ID card was told to buy medicines from private pharmacies.


*I had a seven-month-old sick infant, and I took him to the health facility. They (health workers) examined him and prescribed drugs. I showed them my insurance ID card and told them that the cost should be covered by the scheme. They wrote a prescription and told me to buy the drugs from a private pharmacy. I went back home without having any drugs because I didn’t have money. To me, health insurance is worthless.*
(In-depth interview, woman, age 25 years)

The informants were aware of the maternal waiting homes, despite these facilities’ lacking water, food, and electricity. The women did not prefer to stay there for long when no one was at home to care for their other children.

Furthermore, the interviewed WDG leaders reported that their role was important in improving use of the maternal health services. However, they described that their role was deteriorating from time to time. They expressed that their poor performance negatively affected the use of health facilities. The WDG leaders used to support mothers from all social groups.


*We (women’s development group leaders) are not performing our duties today to the level of women’s satisfaction. We had monthly meetings to monitor the activities of the women’s development groups. In these meetings, we used to report those who were delivering at home and not attending antenatal care. However, today our role is becoming passive due to the weak monitoring from the health extension workers. We have reduced the role we used to play in educating and mobilizing mothers to use the maternal and child health services.*
(In-depth interview, women’s development group leader, age 39 years)

#### 3.3.5. Suggestions to Improve Equity in the Utilization of Health Services

The informants emphasized the importance of improving access to maternal, neonatal, and child health services by moving the services closer to the community. *One woman, aged 22, suggested that there should be a health facility in each tabia in order to enhance equitable access to health services*. Some suggested upgrading health posts to health centers by equipping them with the required health workers, medicines, and other supplies. They also suggested constructing new roads and increasing the number of ambulances. The respondents also highlighted the importance of taking measures against badly behaving health workers and ambulance drivers. It was also suggested to establish self-supporting schemes in the community in order to solve the financial burden of the poor. An interviewed woman suggested establishing a green bank to solve the poor households’ money problem.


*I would suggest contributing cereals, especially during the harvesting season, and store in one place. It is during this season that the farmers easily get grains. If we contributed at least two shember (equivalent to 3 kg) a year, we may solve the financial problem of the poor. By selling the banked cereals, we can cover the poor mothers’ medical and non-medical expenses.*
(In-depth interview, women’s development group leader, age 22 years)

## 4. Discussion

The informants perceived unfair differences in health service utilization, antenatal care, skilled assistance at birth, and services for sick children. Postnatal care was perceived as a missed opportunity. The services that are provided at the fixed health facilities tended to be inequitably distributed [21]. In contrast, child immunization was perceived as an equitable service. This example of equitable access was created by offering this as an outreach service close to home [21]. The informants identified several interrelated social factors that were driving the service inequities; poverty and lack of education made it difficult to reach facilities and pay for medicines. The long distance between homes and health facilities and harsh topography reduced access to services. Culture and traditional norms blocked access to postnatal care. Lack of medicines at health facilities and unfriendly staff experiences prevented mothers from visiting health facilities. The informants perceived the need for a solution to these factors.

During the Millennium Development Goal era, insufficient attention was given to the equity perspective on maternal and child health service utilization. Our study points to a range of interconnected factors behind current inequities. This helps us to understand the local context in order to highlight action areas [22] and address inequities in the utilization of the services. Some of the identified factors are within reach of the health system, and others need action in other sectors.

Ethiopian maternal and child health services are provided free of charge. However, the informants reported that a lack of money for transport, medicines, and food discouraged the poor mothers from utilizing the services. This shows that household poverty is associated with inequity in the utilization of maternal and child health services. Other studies from Ethiopia [23,24] and Sub-Saharan African countries [25] have shown similar results. Poor people lack money for transportation to and from health facilities or to buy pharmaceutical drugs [8]. Out-of-pocket payments may create a financial catastrophe for poor households [11,26] and drive them further into poverty, which may place poor women in an inequitable position in seeking healthcare services [22]. This shows the close connection between the 2030 target for universal health coverage (SDG 3) and the goal for poverty elimination, as defined in SDG 1 [6]. The informants suggested establishing community-based funding schemes to counteract poor women’s financial problems. Empowering poor women through an improved economy could also enhance their healthcare-seeking behavior.

Another important factor brought forward was the lack of education. Uneducated women were more likely to drop out from antenatal care visits and give birth at home. The informants highlighted that the inability to make phone calls was a reason for home delivery among uneducated women. Young women were more educated, and were using maternal and child health services to a greater extent than older women. Similar findings were produced in other studies from Ethiopia [8], which are associated with improved knowledge and understanding of the benefits of MCH services [27]. This is because Ethiopia has created better opportunities for education for the younger generation in the last few decades [28]. These findings link the goal of universal health coverage to access to education, as defined in SDG 4 [29]. Empowering women and girls through education contributes to the enhanced and equitable use of maternal, neonatal, and child health services [14].

A lack of access to health facilities was identified as an impeding factor for poor women in the utilization of maternal and child health services. A long distance from home to health facilities limited the health-seeking behavior of poor mothers from remote areas [25]. This was worsened when the distance was combined with a lack of transport, poor roads condition, and a mountainous topography [24,30]. Similar results were evidenced in a qualitative study from Ethiopia [31] that found distance and a lack of transportation to deter antenatal care visits and skilled assistance at birth. The study also documented that more than 62% of the rural population does not have access to roads where they can obtain transportation. The informants stated that, though ambulances are available to transport the pregnant women for delivery, the uncooperative and unfriendly behavior of drivers makes women give birth at home. This finding was substantiated by a systematic review of low-income African countries [32]. It was evidenced that inequities can be reduced by improving transport services [28].

Additionally, of the socio-cultural factors, a lack of support from husbands and mothers’ responsibility with household chores and childcare were deterrents to antenatal care and facility delivery. The lack of support from husbands was also evidenced in a study from Ethiopia that had negatively impacted women’s health-seeking behavior [31]. These factors have also been reported in other African countries [33]. In a patriarchal society like Ethiopia, gender inequality causes inequities in the use of maternal, newborn, and child health services [14,34]. The socio-cultural construction of gender influences maternal health-seeking behavior [35]. Other studies from Sub-Saharan Africa show similar results [22]. Failure to address gender inequities results in maternal mortality [36]. As such, intervening in improving access to outreach services and promoting gender equity, especially through the WDGs, is likely to have positive consequences for healthcare utilization and health [14,37].

A postnatal visit around the third day after delivery is essential for mothers’ and newborns’ health [38]. However, mothers considered the postnatal visit to health facilities to be limited to sick mothers and neonates, or those who had experienced complications at delivery. This finding is corroborated by a systematic review from Ethiopia [39]. The cultural barriers also hindered mothers from attending postnatal care [40], which was evidenced in an African documentary review showing women’s lack of decision-making [33]. Although Ethiopia introduced home-based postnatal visits, the home visits and counseling were lacking in remote rural settings, resulting in very low coverage of postnatal care [41]. Postnatal care tended to be a missed opportunity to further improve perinatal health. So, the home-based postnatal visists should be strengthened by involving health extension workers and women development group leaders. The cultural norms should be discussed within the networks of women to promote the use of perinatal services.

The informants reported that improper attitudes, maltreatment, and a lack of medicines resulted in a perceived poor quality of services. It was highlighted that women from lower socioeconomic households were more likely to experience maltreatment and abuse by health workers [35,42]. Furthermore, the unavailability of medicines at public health facilities demanded additional out-of-pocket payments [43]. The unfriendly services and unavailability of medicines are evidenced to distance poor mothers from utilizing the otherwise free maternal services [31,39]. Our findings are substantiated by other studies from Ethiopia [44,45] and sub-Saharan African countries [22]. The poor quality services are likely to result in wide health inequities in the utilization of the services [32,46]. As such, bridging the quality gap could reduce inequities in access and the utilization of maternal, neonatal, and child health services [47], with significant implications for the quality of life and well-being of mothers and children [1]. Improved quality of service increases the uptake of maternal health services by more than 50% among all women [39], and reduces both maternal and neonatal deaths by an estimated 28% [46].

Furthermore, inequities in the utilization of maternal and child health services can be explained as insufficiently implemented community-based initiatives, such as health insurance schemes and maternal waiting homes at health facilities. The informants reported that the maternal waiting homes were not suitable for women. This problem was also reported in a study conducted in other parts of Ethiopia [30]. Failure in the implementation of government policies, including insurance schemes in low- and middle-income countries, creates dependence on home delivery [42]. However, a qualitative study from Ethiopia reported that women’s decisions to go to health centers for ANC and delivery are highly influenced by WDGs who work with HEWs. Strengthening community-based interventions would enhance universal health service coverage.

In general, our findings were corroborated with the systematic review performed in low-income countries in Africa [32] and the WHO Commission on the Social Determinants of Health, primarily the structural and intermediary determinants [16]. The various social factors are connected and complex. In Sub-Saharan Africa, inequities in the utilization of maternal and child health services were found in the interactions between these complex natures of the social determinants [36]. Similar evidence has also been documented in a systematic review of low-income African countries [32]. Thus, addressing poverty, access to quality education, and ensuring gender equality will substantially reduce inequities and contribute to universal health coverage [29]. Further studies should assess the interaction of the identified social determinants quantitatively.

### Strengths and Limitations

Our study contributes to the literature on social inequity and perceived factors of inequity in the utilization of maternal, neonatal, and child health services in Ethiopia. We included women from rural communities along with health workers, which allowed us to capture different perspectives and triangulate the evidence. Data were collected from participants in multiple study sites using in-depth interviews and focus group discussions to improve the transferability of the results to other similar settings in Ethiopia. The primary healthcare structure is similar in all of the agrarian rural regions of the country. There was peer debriefing throughout the data collection and analysis in order to ensure the validity of the data. Member checks were also carried out with the health workers. Information collected through different qualitative methods was triangulated in order to enhance credibility. The self-reporting of perceptions and social desirability bias were potential threats to the findings. The study participants were selected by the healthcare providers, which could imply that the mothers hesitated to share sensitive information about the health services. However, we tried to minimize this risk by carefully introducing the study, establishing rapport, and presenting issues with follow-up prompting questions. We cannot generalize the findings because of the small sample size.

## 5. Conclusions

Our informants expressed that maternal health services were perceived to be inequitable in their communities. Child immunization was an exception, due to the outreach model of services. Socioeconomic factors, long distances, and poor-quality health services were identified as leading perceived factors of inequity in the utilization of maternal and child health services. Thus, as part of the efforts to reach the SDGs, equity-oriented policies and interventions need to be integrated with social development programs, such as targeting poverty reduction, improving access to quality education, promoting gender equality and empowerment, and mitigating inequities. These efforts require intersectoral approaches and action in order to reach the SDGs and targets related to maternal and child health.

## Figures and Tables

**Table 1 ijerph-19-11633-t001:** List of major themes and sub-themes that emerged during the analysis.

Major Themes	Sub-Themes	Major Codes
Perceptions about inequity in the use of maternal, neonatal and child health services		
Perceived causes of inequity	Socioeconomic factors	Economic status Education
	Lack of access	Distance and transportation Social and cultural causes Lack of husband support Mother’s heavy housework Mothers cultural taboos
	Poor quality of care	Healthcare providers bad behavior Poor availability of drugs
	Inappropriate implementation of equity-oriented policy initiatives	

## Data Availability

Data from this study are not available. Data anonymity is not obtainable.

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
