# Peer review of "“If I Was Educated, I Would Call the Ambulance and Give Birth at the Health Facility”—A Qualitative Exploratory Study of Inequities in the Utilization of Maternal, Newborn, and Child Health Services in Northern Ethiopia"

_ijerph, 2022, doi:10.3390/ijerph191811633_

Round 1
Reviewer 1 Report
The authors have conducted a qualitative study on perceived inequities in healthcare services for mothers and children among Northern Ethiopian women.
I enjoyed reading the manuscript enriched with live voices by mothers and healthcare workers in Ethiopia, and I think the inequity there should be challenged. However, I regret to say that the manuscript does not deserve publication in IJERPH in its present form.
The interview itself seems well structured, but presentation of the results is very disorganized. The manuscript does not include an overview of the results which would help readers understand the results. Mixed description of responses by mothers and healthcare workers makes readers confused. I cannot get what is the most important or the most annoying for mothers, and what problem the healthcare workers think has the priority to be settled. It means, from a scientific point of view, that an appropriate analysis has not been done, which leads to conclusion without a scientific support.
The authors have made an eye-catching title, but it seems inappropriate because it does not represent conclusions of the present study. Surely, improvement of education would realize more utilization of healthcare service, but it has not been proven by the authors’ study, and there seems other problems including poverty, as the authors themselves concluded.
Though, the manuscript seems suitable for a local journal, to facilitate improvement of the local healthcare system.
Author Response
Dear Editor,
International Journal of Environmental Research and Public Health
We wish to express our appreciation to the reviewers for their comments and suggestions. Their comments have helped us significantly improve our manuscript entitled “If I was educated, I would call the ambulance and give birth at the health facility”. A qualitative exploratory study of inequities in the utilization of maternal, newborn, and child health services in Northern Ethiopia. We appreciate your time and effort in providing valuable feedback on our manuscript. We have incorporated changes in the manuscript reflecting suggestions provided by the reviewers and highlighted these changes within the manuscript. Below is a point-by-point response to the reviewers’ comments.
Sincerely yours
For the group of authors
Alem Desta Wuneh
A point-by-point response to Reviewer 1
The authors have conducted a qualitative study on perceived inequities in healthcare services for mothers and children among Northern Ethiopian women.
I enjoyed reading the manuscript enriched with live voices by mothers and healthcare workers in Ethiopia, and I think the inequity there should be challenged. However, I regret to say that the manuscript does not deserve publication in IJERPH in its present form.
The interview itself seems well structured, but presentation of the results is very disorganized. The manuscript does not include an overview of the results which would help readers understand the results. Mixed description of responses by mothers and healthcare workers makes readers confused. I cannot get what is the most important or the most annoying for mothers, and what problem the healthcare workers think has the priority to be settled. It means, from a scientific point of view, that an appropriate analysis has not been done, which leads to conclusion without a scientific support.
Response: We appreciate that the reviewer enjoyed the manuscript, but regret that the reviewer felt that results were disorganized. Contrary to what the reviewer states, the Result section starts with an overview of main themes and sub-themes in results, see Table 1 page page 4. The structure presented in this table is pursued throughout the Results section. Mothers’ and healthcare workers’ perceptions were triangulated and contrasted to show where lived experiences and opinions were shared or not shared by the two groups. Important issues were illustrated by quotations. We disagree with the opinion expressed by the reviewer that we should have failed pursuing an appropriate analysis of the qualitative data.
The authors have made an eye-catching title, but it seems inappropriate because it does not represent conclusions of the present study. Surely, improvement of education would realize more utilization of healthcare service, but it has not been proven by the authors’ study, and there seem other problems including poverty, as the authors themselves concluded.
Response: As reflected in a quotation from a health worker, the problem of low utilization of health services is multifaceted. However, as also shown in the results, the enhanced education level among younger mothers makes a difference in spite of persistent poverty and distance to services. As commented in the Conclusion, reaching the non-health-related goals, such as education, is essential also for universal health coverage and other health-related SDG targets. We consider the brief education quotation in the manuscript’s title to represent one of the most important conclusions and would prefer not to change the title.

Reviewer 2 Report
I think the paper is well-written, but a few points need to be clarified.
1. (Title) Why did you include it in the tile? It is rate to include these kind of statement in title.
2. (Introduction) Why did you conduct a qualitative study instead of quantitative study? It was probably possible to conduct a quantitative study, and the results of quantitative study would be more appealing. It is better to written a reason in Introduction or Discussion.
3. (Abstract & line 84) What does “focus group discussion” mean?
4. (Limitation) It is written in the limitation that the findings of this study cannot be generalizable because of small sample size. In this case, significance of the results is uncertain.
Author Response
Dear Editor,
International Journal of Environmental Research and Public Health
We wish to express our appreciation to the reviewers for their comments and suggestions, which have helped us significantly improve our manuscript entitled “If I was educated, I would call the ambulance and give birth at the health facility”. A qualitative exploratory study of inequities in the utilization of maternal, newborn, and child health services in Northern Ethiopia. We appreciate your time and effort in providing valuable feedback on our manuscript. We have incorporated changes in the manuscript reflecting suggestions provided by the reviewers and highlighted these changes within the manuscript. Below is a point-by-point response to the reviewers’ comments.
Sincerely yours
For the group of authors
Alem Desta Wuneh
A point-by-point response to Reviewer 2
I think the paper is well-written, but a few points need to be clarified.
- (Title) Why did you include it in the tile? It is rare to include these kinds of statement in title.
Response: We consider this quotation to represent an important result and conclusion. It is relatively common to include a brief quotation in the title of a qualitative paper.
- (Introduction) Why did you conduct a qualitative study instead of quantitative study? It was probably possible to conduct a quantitative study, and the results of quantitative study would be more appealing. It is better to written a reason in Introduction or Discussion.
Response: We have earlier conducted a quantitative study on inequity in the utilization of maternal and child health services. By a qualitative approach, we get the opportunity to gain knowledge of mothers’ and health workers’ lived experiences and perceptions of inequity in the utilization of primary health services. We refer to the earlier quantitative publication in the Introduction.
- (Abstract & line 84) What does “focus group discussion” mean?
Response: Focus group discussion is a qualitative method that collects data through group interaction on the study topic, i.e., perception of inequity in the utilization of maternal and child health services. In this study, three focus group discussions were conducted.
- (Limitation) It is written in the limitation that the findings of this study cannot be generalizable because of small sample size. In this case, significance of the results is uncertain.
Response: Yes, the sample size is small, which is a typical feature of qualitative research. This is because the aim of qualitative research is not the generalizability, rather the in-depth understanding of the issue under investigation. So, the significance of the results in qualitative research is not tested using sample size rather the trustworthiness of the information. The study was conducted in multiple sites involving women and health care workers. The description of the results is thick, enhancing transferability of the results to similar settings. As usual in qualitative studies, the sample size was based on saturation in the collection of data.
